# Paracetamol: A Review of Guideline Recommendations

**DOI:** 10.3390/jcm10153420

**Published:** 2021-07-31

**Authors:** Ulderico Freo, Chiara Ruocco, Alessandra Valerio, Irene Scagnol, Enzo Nisoli

**Affiliations:** 1Anesthesiology and Intensive Care, Department of Medicine—*DIMED*, University of Padua, 35122 Padua, Italy; irene.scagnol.3@gmail.com; 2Center for the Study and Research on Obesity, Department of Biomedical Technology and Translational Medicine, University of Milan, 20129 Milan, Italy; chiararuocco@gmail.com (C.R.); enzo.nisoli@unimi.it (E.N.); 3Department of Molecular and Translational Medicine, University of Brescia, 25100 Brescia, Italy; alessandra.valerio@unibs.it

**Keywords:** pain, musculoskeletal, cancer, headache, elderly, paracetamol, guidelines

## Abstract

Musculoskeletal pain conditions are age-related, leading contributors to chronic pain and pain-related disability, which are expected to rise with the rapid global population aging. Current medical treatments provide only partial relief. Furthermore, non-steroidal anti-inflammatory drugs (NSAIDs) and opioids are effective in young and otherwise healthy individuals but are often contraindicated in elderly and frail patients. As a result of its favorable safety and tolerability record, paracetamol has long been the most common drug for treating pain. Strikingly, recent reports questioned its therapeutic value and safety. This review aims to present guideline recommendations. Paracetamol has been assessed in different conditions and demonstrated therapeutic efficacy on both acute and chronic pain. It is active as a single agent and is additive or synergistic with NSAIDs and opioids, improving their efficacy and safety. However, a lack of significant efficacy and hepatic toxicity have also been reported. Fast dissolving formulations of paracetamol provide superior and more extended pain relief that is similar to intravenous paracetamol. A dose reduction is recommended in patients with liver disease or malnourished. Genotyping may improve efficacy and safety. Within the current trend toward the minimization of opioid analgesia, it is consistently included in multimodal, non-opioid, or opioid-sparing therapies. Paracetamol is being recommended by guidelines as a first or second-line drug for acute pain and chronic pain, especially for patients with limited therapeutic options and for the elderly.

## 1. Introduction

Musculoskeletal pain (MSP) conditions are the main contributors of years lived with disability worldwide [1,2,3]. They include back and neck pain, hip, and knee osteoarthritis (OA), rheumatoid arthritis, gout, and a heterogeneous group of autoimmune, inflammatory, and degenerative disorders of joints, tendons, and muscles. Acute and chronic pain, stiffness, and impairment of personal and social activities and of quality of life are unifying features [1,2,3].

Most MSP conditions increase with age. As a result of the rapid aging of the world population, the global impact of MSP and of related disability will steadily and markedly increase in the near future [1,3]. Available pharmacological treatments with non-steroidal anti-inflammatory drugs (NSAIDs) with or without opioids improve MSP in young and/or otherwise healthy people; however, these drugs are often useless in the elderly, frail, or sick patients because of comorbidities and contraindications. NSAIDs are commonly prescribed in the setting of acute MSP. Medical societies, including the American Geriatric Society, the American College of Rheumatology, and the European League Against Rheumatism, recommend extreme caution when giving NSAIDs to the elderly and limit their use to the lowest effective dose and the shortest duration [4]. Since even a short course with NSAIDs has been associated with severe adverse events, gastro-intestinal, renal, and cardiovascular side effects should be routinely monitored [4,5]. As NSAIDs are not indicated for chronic use, opioids have been the mainstay for the long-term treatment of chronic MSP [6,7,8]. However, in contrast to positive reports in young patients, there is a lack of well-designed specific studies on the efficacy and safety of opioids in the elderly patients [6,7,8]. Furthermore, when opioids are used in the elderly, the age-related physiological decline of hepatic and renal functions require a slow titration and frequent monitoring for potential adverse events [6,7,8]. Cardiovascular and respiratory disorders with the risk of respiratory depression are further complications that require careful dosing and often prevent achieving an adequate analgesia in a substantial percentage of patients [6,7,8]. In addition, the abuse and misuse of opioids have caused enormous economic and social costs of the so-called opioid epidemic and led to the revision of guidelines and recommendations against their use in MSP [9,10]. Finally, the development of new drugs for pain has been plagued by failures in advanced human trials, leaving clinicians with few therapeutic options to treat pain.

Paracetamol is one of the most used drugs both over the counter and on prescription for pain and fever [11]. It has a unique clinical pharmacological profile that includes potent analgesic and antipyretic effects and no or little anti-inflammatory activity as well as minor gastrointestinal, renal, and vascular side effects (Table 1). For a long time, paracetamol has been recommended as a first-line drug in pain management guidelines. Recently it underwent intense investigations with reports showing that its analgesic efficacy may be lower than previously thought. Pharmacoepidemiology and pharmacovigilance studies report the occurrence of acute liver injury in association with paracetamol utilization.

However, because of its good safety record, paracetamol remains a recommended analgesic, especially for aged and frail patients. Furthermore, its efficacy is enhanced in fast-dissolving formulations, and it has a useful opioid-sparing activity that reduces adverse events and risks from high doses of opioids. The objective of this paper is to conduct a scoping literature to summarize the evidence and the guideline recommendations on paracetamol for pain.

## 2. Methods

This is a scoping review aiming to provide an overview of the current guidelines on paracetamol for the management of most common pain conditions. The review was guided by the methodological framework devised by Arksey and O’Malley and subsequently modified by the Johanna Briggs Institute [12]. The PRISMA Extension for Scoping Reviews (PRISMA-ScR) was followed to summarize the screening methods of the review (Figure 1) [13].

Relevant reports of guidelines were identified by searching the CINAHL, Cochrane EMBASE, and Medline databases utilizing the following strategy. The words “paracetamol/acetaminophen” were explored with the words “pain”, “randomized controlled trial” (RCT), “review”, “meta-analysis”, and “guidelines”. Two authors (UF and IS) independently screened abstracts and papers; in case of disagreement, a third author was consulted (EN). The criteria for including studies into this review were as follows: to be claimed as guidelines; to be authored by a specific health organization or medical society; to report a detailed methodology including the definition of the target population, data selection, methods for decision making, and the specific aims of the guidelines; to deal with one of the following pain conditions: MSP, cancer pain, and headache; to be published between 2000 and 2021. Exclusion criteria were as follows: studies not conducted to develop guidelines; guidelines dealing with other pain conditions (i.e., dysmenorrhea, dental pain, ear pain, eye pain, and pain in the neonatal or pediatric population); guidelines written in a language other than English.

The AGREE II instrument [14] was used to determine the methodological quality of the included guidelines in six domains: scope and purpose, stakeholder involvement, rigor of development, clarity and presentation, applicability, editorial independence, and an overall assessment. The 23-item AGREE II instrument uses a 7-point agreement scale from 1 (strongly disagree) to 7 (strongly agree). Each guideline was independently scored by two authors (UF and IS). In case of a significant discrepancy, a third author was consulted (EN). Each item was scored, and a total quality score was calculated for each domain by summing the score of each item. The mean domain scores between the two raters was used to standardize the domain score as a percentage.

## 3. Paracetamol

### 3.1. Acetaminophen or Paracetamol

Acetaminophen (paracetamol; N-acetyl-p-aminophenol) is the active metabolite of phenacetin. Unlike phenacetin, paracetamol is not carcinogenic. It is available on prescription and over-the-counter as a widely used reliever of fever and pain [11,15,16,17,18,19]. Paracetamol is well-tolerated and safe, without several of the side effects typically observed with aspirin. During the middle- and late-19th century, pharmacologists isolated salicin and salicylic acid. The French chemist Charles Frederic Gerhardt (Strasburg, 1816–1856) and the Bayer chemist Felix Hoffmann (Ludwigsburg, 1868–Switzerland, 1946) developed the synthesis methods to produce acetylsalicylic acid. In the 1880s, the cinchona tree became limited, and production alternatives were sought. Acetanilide was synthesized in 1886, and phenacetin was synthesized in 1887. In 1878, Harmon Northrop Morse (1848–1920) synthesized paracetamol by reducing p-nitrophenol with tin in glacial acetic acid. Still, this substance only became widely used as a drug after Morse’s death. Interestingly, paracetamol was found in the urine of subjects consuming phenacetin, and it was discovered as a metabolite of acetanilide in 1899, yet the finding was still ignored at the time. In 1946, Bernard Brodie (Liverpool, 1907–Charlottesville, 1989) and Julius Axelrod (New York, 1912–Rockville, 2004) investigated why non-aspirin agents were related to the development of methemoglobinemia, which is a disease that decreased the oxygen-carrying capacity of blood and that was potentially lethal. In 1948, Brodie and Axelrod explained that acetanilide caused methemoglobinemia. Then, they determined that acetanilide’s analgesic effect was due to its active metabolite paracetamol and that paracetamol had no toxic effects on acetanilide [15,16,17,18,19].

The product was first sold in 1955 by McNeil Laboratories as a pain and fever reliever for children, under the brand name Tylenol Children’s Elixir. Paracetamol 500 mg tablets went on sale in the United Kingdom (Panadol) in 1956 and were initially available only by prescription, and it was marketed for the treatment of pain and fever. In contrast to other analgesic agents containing aspirin, paracetamol was not a stomach irritant. Paracetamol was added to the British Pharmacopoeia in 1963 and has gained popularity since then as an analgesic agent with few side effects and little interaction with other pharmaceutical agents.

### 3.2. Chemistry

Paracetamol is constituted by a benzene ring core substituted by one hydroxyl group and the nitrogen atom of an amide group (acetamide) in the para (1,4) pattern (Figure 2).

The molecule is extensively conjugated: the lone pair on the hydroxyl oxygen, the benzene pi cloud, the nitrogen lone pair, the p orbital on the carbonyl carbon, and the lone pair on the carbonyl oxygen are all conjugated. The benzene ring is highly reactive toward electrophilic aromatic substitution by the presence of two activating groups. This conjugation markedly diminishes the basic value of oxygen and nitrogen atoms, while making acid the hydroxyl group through delocalization of the charge developed on the phenoxide anion.

### 3.3. Mechanisms of Actions 

The mechanisms of the analgesic activity of paracetamol are not fully understood and may involve Peripheral and Central Nervous System sites of action [15,16,17,18,19,20,21,22,23,24,25,26,27,28,29,30]. It is widely accepted that paracetamol decreases the tissue concentrations of prostaglandins and pro-inflammatory mediators, whose synthesis is also inhibited by aspirin (acetylsalicylic acid). However, unlike aspirin, paracetamol is not featured by a significant anti-inflammatory activity and does not inhibit the synthesis of pro-clotting thromboxanes. Although it can inhibit cyclooxygenase (COX) enzymes, paracetamol may act via two major alternative molecular pathways [15,16,17,18,19,20]. The prostaglandin G/H synthase enzymes, also known as the COXs, function as essential enzymes for metabolism of the arachidonic acid metabolism to prostaglandin G/H, which is an unstable molecule quickly converted to other pro-inflammatory derivatives. NSAIDs selectively block this step. There are two COX forms, COX-1 and COX-2. Inhibition of COX-2 is thought to mediate the antipyretic, analgesic, and anti-inflammatory actions of NSAIDs. Aspirin is a non-competitive, irreversible inhibitor because it acetylates the isozymes in the aspirin-binding channel. Paracetamol acts as a non-competitive reversible inhibitor by reducing the peroxide site of the enzymes [16,19].

Paracetamol can affect the central neurotransmission of pain in different ways [18,20,21,22,23,24,25,26,27,28,29,30]. In particular, the drug is metabolized to N-arachidonoylaminophenol (AM404), which is a compound with multiple potential analgesic activities, including the blockade of neuronal uptake of anandamide and of neuronal sodium channels [20,23,26].

Cholinergic, noradrenergic, opioid, and serotoninergic (5-HT) mechanisms are thought to be involved in the complex “central”, spinal and supraspinal, actions of paracetamol [18]. In experiment animals, the blockade of the 5-HT neurotransmission by the neurotoxic lesion of descending 5-HT pathways, inhibition of 5-HT synthesis, or antagonism of 5-HT_3_ receptors reverses paracetamol antinociception in rodents [18]. Ondansetron and tropisetron, two 5-HT_3_ antagonists, abolish paracetamol analgesia in humans [23,24]. Naloxone and naltrexone, two μ-opioid receptor antagonists, reduce or abo-lish paracetamol analgesia in different animal pain models [25,26].

Pickering et al. investigated the central antinociceptive effects of paracetamol in humans [27,28]. Using the blood oxygenation level-dependent signal of the functional Magnetic Resonance Imaging, they measured the cerebral blood flow responses to a thermal stimulus of moderate-to-severe intensity (i.e., pain Numerical Rating Scale (NRS) 6/10) in healthy volunteers receiving placebo or paracetamol in randomized, double-blind, crossover design sessions. Compared to the placebo sessions, in the paracetamol sessions, the pain-induced increases of cerebral blood flow were significantly attenuated in the prefrontal cortices, insula, thalamus, anterior cingulate cortex, and periaqueductal gray matter [27]. The findings suggest that paracetamol negatively modulates the ascending spinothalamic projection to supraspinal cortical areas.

Although, opioids do not alter cognition in chronic pain patients when given at stable doses, their potential cognitive impact is a concern, especially in the elderly population [6,7]. In contrast, Pickering et al. also showed that paracetamol positively affects cognition in decision making and working memory domains [28]. Cognitive performances were measured in 40 healthy volunteers 1 week apart with a set of cognitive tests (i.e., information sampling task for pre-decisional processing, Stockings of Cambridge for spatial memory, reaction time, delayed matching of a sample, and pattern recognition memory) before and after random oral administration of placebo or 2 g paracetamol. Treatment with paracetamol improved tasks of information sampling, spatial planning, and working memory [28]. The implications are twofold: firstly, the findings confirmed the clinically relevant effects paracetamol has on the Central Nervous System; secondly, paracetamol has positive effects on cognition [28].

### 3.4. Pharmacokinetics

Oral paracetamol has excellent bioavailability with peak plasma concentrations occurring within 30–60 min after ingestion; the plasma t_1/2_ is about 120 min. Its binds to plasma proteins less than NSAIDs and diffuses throughout most body fluids. The kidneys excrete glucuronide conjugates. Some 90–100% of the drug may be recovered in the urine within the first day at therapeutic dosing [18,19]. Paracetamol is principally transformed into inactive compounds through the conjugation with sulfate and glucuronide, and a small portion is oxidized via the cytochrome P450 enzyme system (its CYP2E1 and CYP1A2 isoenzymes). The CYP2E1 and CYP1A2 convert paracetamol into the alkylating metabolite (N-acetyl-p-benzoquinone imine, NAPQI), which may be responsible for paracetamol liver toxicity (see also Section 6. Safety and Toxicity). Based on the levels of CYP2D6 expression, individuals can be classified into “extensive”, “ultrarapid”, and “poor metabolizers” [30].

### 3.5. Oral vs. Intravenous Formulations

Paracetamol is available in tablets, suppositories, and oral and injectable solutions. The standard adult dose is 500 to 1000 mg, while adult’s recommended maximum daily dose is 3 to 4 g. In the last two decades, intravenous and oral rapidly dissolving preparations, granules, or tablets became widely available. They provide faster T_max_ and higher C_max_ than tablets with the oral solution achieving higher bioavailability than intravenous formulation [31]. Intravenous paracetamol produces peak plasma concentrations in approximately 15 min compared to 45–50 min following oral administration, resulting, theoretically, in a faster onset of the analgesic effect (5 min) [31,32]. However, several studies, demonstrate similar analgesic efficacy of intravenous and oral preparations [31,32,33,34,35,36]. In a double-blind RCT, a heterogeneous group of 87 patients of the Emergency Department with moderate-to-severe pain (median age of 45 years, 60% females) were randomized to receive 1 g of paracetamol either intravenously or orally; the changes in Visual Analogue Score (VAS) for pain from baseline (67.9 ± 16.0 mm) to 30 min post-administration outcome did not differ between groups (−16.0 ± 19.1 mm in the intravenous group and −14.6 ± 26.4 in the oral group, p = 0.79) [33]. Secondary outcomes including the length of stay, patient satisfaction, and need for rescue analgesia did not differ between groups [33]. The authors concluded that intravenous and oral paracetamol produced a small but clinically significant decrease in pain [33]. In an RCT comparing intravenous and oral paracetamol in 120 patients undergoing hip and knee arthroplasty, 1 g of intravenous or oral paracetamol was administered preoperatively and then every 6 h for 24 h. The 24 h average pain VAS and 24 h hydromorphone rescue analgesia did not differ between the two groups, except for a lower pain VAS in the intravenous group at the postoperative 0–4 h interval [32]. In a double-blind RCT, 154 patients undergoing a total hip arthroplasty procedure received either intravenous or oral 1 g of paracetamol as part of a postoperative opioid-sparing, multimodal analgesia (i.e., 15–30 mg intravenous ketorolac every 8 h, for a total of 6 doses, and then oral meloxicam 7.5–15 mg until postoperative day three or discharge; upon request, tramadol 50–100 mg for mild–moderate pain or oxycodone 5 mg for severe pain; intravenous 2 mg hydromorphone as rescue analgesia for severe pain). Paracetamol was administered 30 min after admission to the post-anesthesia care unit and then every 6 h for three days or to discharge [34]. The pain NRS during physical therapy on postoperative day 1 was similar in the intravenous and oral treatment group (3.9 ± 2.4 and 3.6 ± 2.4); the cumulative doses of oral morphine equivalent were also similar between groups [34]. The authors concluded that patients in both groups had low pain scores and limited opioid usage [34]. Much alike, Johnson et al. found that a single preoperative administration of 1 g of oral paracetamol produced postoperative analgesia similar to 1 g of intravenous paracetamol in patients undergoing laparoscopic cholecystectomy [32]. Fenlon et al. studied 130 patients treated with oral or intravenous paracetamol and oral or intravenous placebo for third molar extraction; there was no difference in the analgesic outcome of satisfactory analgesia at 1 postoperative h [35]. In a meta-analysis of six RCTs comparing intravenous versus oral administration, Jibril et al. found no evidence indicating that the increased bioavailability of the intravenous preparations produces a superior analgesia [31]. In a recent systematic review of 14 trials with 1695 participants on postoperative pain, there was no significant difference between intravenous and oral paracetamol in terms of pain intensity up to >24 postoperative h [36].

## 4. Pain

Pain is a health problem of epidemic proportions with 15% and 25% of people reporting to suffer from pain most or every day over the last 3 to 6 months; pain increases with age and with low socioeconomic status [1,2,3,4]. Persistent, intense pain can impair a person’s mental and physical well-being. For these reasons, in 1995 James Campbell, in his Presidential Address to the American Pain Society, proposed that pain should be measured as a fifth vital sign, along with blood pressure, temperature, heart rate, and respiratory rate. While the initiative was intended to improve pain care, the lack of long-term safe treatments of pain led to increased opioid prescriptions that eventually contributed to the opioid crisis [37]. As a result of the large numbers of deaths from overdose of prescribed opioids, in 2016, the American Medical Association voted to stop considering pain as the fifth vital sign, to reduce opioid prescription, and to shift to non-opioid therapies to manage pain [38].

Acute and chronic pain are different clinical entities. Acute pain is viewed as a physiological, time-limited, protective response to a specific injury that resolves with healing. In contrast, chronic pain may be considered a ‘*disease state in its own right*’ according to the European Pain Federation. It outlasts the expected time of healing from a disease or injury, has no biological purpose and, often, no recognizable cause. OA is the most frequent cause of chronic pain with a prevalence varying markedly depending on age range considered, gender and geographic distribution, genetics and lifestyles, and the method has defined. Lumbar spine OA is the single leading cause of disability, with estimates ranging from 40% to 85% [39,40]. Large joint OA is most common in the knee, followed by the hand and hip, affecting 10% of men and 13% of women aged 60 years or older [40]. Radiographic investigations reveal an earlier and larger prevalence for asymptomatic radiographic OA than for symptomatic OA [40].

Genetic factors and female sex represent 40% to 80% of cases in hand and hip OA cases while accounting for somewhat less in knee OA. Obesity is a risk factor for knee OA and joint deformity is a risk factor for hip OA. The several environmental risk factors for lower limb OA include joint injury from high-impact sports and heavy work activities involving lifting, cumulative physical loads, full-body vibration, and bending/kneeling/squatting [41]. However, the most relevant risk factor of OA is aging because of age-related cumulative exposure to risk factors and degenerative changes in joint structures. Prevalence of knee and hand OA rises more rapidly in women than in men after the age of 50 years, peaking at the age of 75 years. MSP from multiple joint sites is a common occurrence [42]. According to the Osteoarthritis Foundation, currently, 300 million mostly older people may suffer from OA worldwide; these numbers are expected to rise because of the rapid aging and the increasing obesity in the global population.

OA is primarily a degenerative joint disease characterized by cartilage damage and remodeling and inflammation of joint structures [43]. In the elderly, OA is almost constantly associated with sarcopenia and tendinopathies that both worsen joint stability and pathology [44]. OA pain is mostly a nociceptive pain that arises from peripheral nociceptors stimulated by movement and/or inflammatory reactions; then, it is transmitted through myelinated Aβ and Aδ fibers and unmyelinated and C fibers to the spinal cord and then through the spinothalamic tracts to supraspinal, pain-processing cortical regions. The cartilage is physiologically aneural and cannot directly cause pain; however, it can become pathologically innervated and, along with densely innervated subchondral bone, synovium, and joint structure, can generate intense pain [43]. A neuropathic pain component resulting from the pathological innervation of cartilage and/or central sensitization is a common occurrence in OA [43]. OA causes a considerable socioeconomic burden as it leads to loss of productive days and years in adulthood and to loss of Quality of Life during aging. Furthermore, lower limb OA has been positively associated also with increased cardiovascular mortality probably, in part, via unrefreshing sleep, depression, and reduced activity [45,46]. However, OA-related pain and disability are often underdiagnosed and undertreated, especially in the elderly; for example, 72% of hip fracture patients receive no prehospital treatment for pain [47]. Patients and their caregivers often regard pain as a common natural occurrence of aging or because elderly patients are thought to feel less pain, be cognitively impaired, or be reluctant to report pain for stoicism [47,48,49]. A negative attitude toward treating pain in the elderly is favored by a lack of studies on the efficacy and safety of pharmacological treatments of OA pain in the elderly [50,51,52,53,54,55,56]. In a review of 83 clinical studies involving >10,000 subjects treated with analgesics, only 2.3% of people were over 65 years [50]. Furthermore, aging is the most critical risk factor for cardiovascular disease, and elderly OA patients have one of the highest multimorbidity rates in general practice [52,53,54]. In Australian studies, older patients with OA also had hypertension (>50%), cardiovascular disease (20%), dyslipidemia (19%), diabetes (14%), and mental health disorders (12%), many of which are contraindications to NSAIDs and opioids practice [55]. In diabetic patients, although it is under prescribed, the prevalence of NSAIDs contraindicating antiplatelet therapy is >50% [56]. Side effects are common and may lead to treatment discontinuation.

Cancer is a leading cause of morbidity and mortality, with more than 18 million new cases and 9 million deaths in 2018 globally [57]. Pain is frequent in cancer, affecting approximately 55% of patients undergoing anticancer treatment, and 66% of patients with advanced or terminal disease [57]. Pain can be nociceptive arising from direct tissue invasion and/or from a perilesional inflammatory reaction; neuropathic pain is caused by nerve involvement from cancer- or therapy-related nerve injury. Treatment can be challenging because patients are often older, frail, highly comorbid, or with multiple and/or end-stage organ failure [57,58]. However, in cancer patients, early pain therapy is critical not only to reduce pain discomfort but slow the disease progression and improve survival [58].

## 5. Clinical Guidelines

Medical institutions and societies develop clinical practice guidelines to ensure the best treatment to patients. Expert panels draw them up with regarding the pertinent literature and a consensus opinion is reached. Panelists grade the quality of evidence in a rather standard fashion, most frequently from I to V with I indicating high-quality evidence from properly designed RCTs and meta-analysis and V indicating lower quality evidence from case reports and expert opinions. The strength of treatment recommendations is graded on the basis of quantity and quality of the evidence most frequently from A to D (A: high quality of evidence; effective. B: mode-rate quality of evidence; probably effective. C: low quality of evidence; probably ineffective. D: very low quality of evidence; not effective). The GRADE (Grading of Recommendations, Assessment, Development, and Evaluations) approach is frequently used [59]. However, the panelists take into account not only the clinical research findings collated through systematic reviews and meta-analyses on treatment efficacy but also the specific definitions of clinical conditions, the benefit/harm balance, the economic costs, the epidemiological context, the availability of alternative treatments, the patient acceptability, and their own clinical experience. The agreement of panelists is obtained in different ways. As a consequence, the analyses of the same evidence may lead to different recommendations by different committees. Clinical guidelines are updated periodically to keep up with new scientific discoveries, new treatments, and emerging pharmacoepidemiological data. The efficacy of paracetamol has been determined in RCTs as well as real-life studies to show its superiority versus placebo or other pain drugs [33,34,35,36,60,61,62,63,64,65,66,67,68,69,70]. However, recently, paracetamol failed to show superiority over placebo, and questions have been raised about its safety [70]. However, at the same time, severe concerns have been raised about the organ toxicity by NSAIDs particularly in the elderly and the risk of abuse and overdose of opioids [4,5,9,10]. The opioid epidemic is still underway. As a consequence, paracetamol is still maintained or included in clinical guidelines.

The search strategy identified 716 publications with 109 duplicates. After screening and review, 17 documents containing 18 guidelines were included in the review (Figure 1 shows the PRISMA flow chart).

The AGREE II domain scores for each guideline are displayed in Table 2. The mean scores for each domain were (range ± SD): scope and purpose 96.1 ± 4.6; stakeholder involvement 73.5 ± 14.8; rigor of development 64.5 ± 17.0; clarity of presentation 90.6 ± 9.4; applicability 26.1 ± 21.4; editorial independence 65.8 ± 22.6. On the overall assessment, 10 guidelines were judged as recommended since their quality scores was 5 to 7, representing high- or good-quality guidelines; 7 guidelines were recommended after modification (quality scores 3 and 4); 1 guideline was scored 2 and not recommended.

### 5.1. Acute Pain

Acute pain requires an analgesic with a fast onset of action. Fast dissolving tablets and intravenous and oral solutions of paracetamol have been developed for this purpose. Paracetamol can be administered alone to treat mild-to-moderate pain [33,34]. Paracetamol is often co-administered with NSAIDs or opioids with remarkable drug-sparing effects in subjects with severe pain [33,34,65].

Hung et al. randomized 783 patients, with a soft tissue injury, to receive either paracetamol 1000 mg or ibuprofen 400 mg or paracetamol 1000 mg and ibuprofen 400 mg. The pain intensity on the 0–100 mm VAS pain scale declined by 12, 12, and 13 mm, respectively, in the paracetamol, ibuprofen, and in the combined paracetamol and ibuprofen groups. The authors concluded that the treatments were clinically effective without significant differences between groups [60].

Paracetamol was not inferior to diclofenac or indomethacin in the management of acute MSP [61,62,63]. A Cochrane review on nine studies with 991 patients, comparing paracetamol with NSAIDs for acute soft tissue injury, found no clinically relevant differences (low-to-moderate quality evidence) between patients treated with paracetamol or with NSAIDs in pain-attenuating effects at day 1–7, swelling, and return to function at day 7; gastrointestinal adverse events were more common in patients treated with NSAIDs [64]. In a qualitative review of RCTs comparing paracetamol with NSAIDs, Hyllested et al. found that while it was less effective than NSAIDs in dental surgery, paracetamol was equally effective to NSAIDs in major surgery and orthopedic surgery; also, paracetamol enhanced analgesia when added to a NSAID, compared with NSAIDs alone [65]. The authors concluded that paracetamol is a viable alternative to NSAIDs because of the low incidence of adverse effects, and it should be the preferred analgesia in high-risk patients [65]. Although less active on the severe hip or knee OA, paracetamol was equally effective than diclofenac/misoprostol in mild OA [66].

A large RCT involving 1644 adult patients with acute renal colic compared the anal-gesic efficacy of 1000 mg intravenous paracetamol, 75 mg intramuscular diclofenac, or 0.1 mg/kg intravenous morphine intravenously [67]. The primary analgesic outcome of 50% pain reduction at 30 min post-administration was achieved by 66%, 68%, and 61% with no differences among groups [67]. In a prospective cohort study in 116 patients presented to the Emergency Department of a level one trauma center because of fractures, strains, or sprains receiving opioids in the ambulance or during their stay in the Emergency Department, paracetamol did not modify morphine requirement in the acute phase or after discharge [68]. In a recent systematic review, paracetamol was found to be as effective as NSAIDs in treating acute MSP in patients with minor musculoskeletal injuries in terms of analgesic efficacy need for additional analgesia and adverse events (low quality of evidence) [69].

Multimodal analgesia is recommended for severe postoperative or trauma pain [71]; it is based on the premise that the combined use of different analgesic drugs and technique primarily non-opioid analgesics that can have additive or synergistic effects that provide superior analgesia while reducing opioid dosing and related adverse effects. Paracetamol has a role in the multimodal approach [71]. Miranda et al. used isobolographic analysis to calculate the effects of paracetamol on the ED_50_ of different NSAIDs in mice; they demonstrate that all the combinations were synergistic, the experimental ED_50s_ being significantly smaller than the theoretically calculated ED_50s_ [72]. Zeidan et al. formally determined the median ED_50s_ of paracetamol and morphine alone and paracetamol and morphine combination using the Dixon and Mood up-and-down method in three groups of 30 patients undergoing moderately painful surgery; initial doses were 1.5 g and 5 mg in the paracetamol and morphine groups, and they were 1.5 g and 3 mg in the paracetamol–morphine combination group [73]. The median ED_50s_ of paracetamol and morphine alone were 2.1 g and 5 mg in paracetamol and morphine groups, respectively, and they were 1.3 g for paracetamol and 2.7 mg for morphine in the combination treatment group. The results demonstrated that paracetamol and morphine share, at least, additive analgesic effects [73]. In elderly painful conditions, the non-opioid analgesics such as paracetamol should be continued to facilitate “opioid-sparing” dosing [74]. A systematic review of the peer-reviewed literature confirmed that the combined use of paracetamol and NSAIDs could significantly enhance the analgesic effects of either drug alone [75,76]. Furthermore, in patients with rheumatoid arthritis, the co-administration of a fish oil containing the n-3 fatty acid, eicosapentaenoic acid enhanced the inhibition of COX-2 generated prostaglandin E2 [77].

Paracetamol has been used for a long time for the treatment of headache and migraine. Using the American Academy of Neurology criteria to develop guidelines, the American Headache Society considered oral paracetamol effective with a level of evidence A (established as effective) when it is used alone or in combination with aspirin for non-incapacitating attacks of migraine (effective), with a level B (probably effective) when is used in combination with codeine or tramadol, and with a level C (possibly effective) when used in combination with butalbital [78]. In an RCT, 1 g of oral paracetamol was superior to placebo in terms of 2 h headache relief (i.e., responder percent after paracetamol or placebo 51% vs. 27%, P = 0.008) and of associated symptoms such as functional disability, photophobia, and phonophobia [79]. Not all studies were positive. In fact, in contrast to previous reports, intravenous 1 g of paracetamol for treating an acute migraine attack failed to demonstrate significant differences over placebo in terms of headache freedom and relief at two and at 24 h (31% vs. 33% placebo; P = not significant) [80]. However, paracetamol is recommended for the treatment of migraine by most national and international societies including the American Headache Society, the American Academy of Fa-mily Physicians and the American College of Physicians–American Society of Internal Medicine, and the Ad Hoc Committee of the Italian Society for the Study of Headaches for the Guidelines of Primary Headaches in adults (Table 3) [78,81,82]. The European Federation of Neurological Societies supports paracetamol as first-line treatment a high degree of evidence (level A) for tension-type headache [83].

### 5.2. Chronic Pain

The management of recurrent and chronic pain requires a stepwise approach with an initial recommended treatment in guidelines with paracetamol and topical agents [9,84]. In recent years, several studies and meta-analyses have compared the analgesic efficacy and safety of paracetamol versus placebo or NSAIDs for chronic pain. Some evidence of reduced efficacy of paracetamol has changed some opinions and guidelines. Nevertheless, paracetamol continues to be used, clinically tested, and recommended by several clinical practice guidelines that were based on expert opinions and structured reviews of the literature (Table 2).

Five hundred and seventy-one patients with hip or knee OA were randomly treated for 6 or 12 months with 4 g/day paracetamol or naproxen 750 mg/day. Paracetamol improved the Western Ontario and McMaster Universities Osteoarthritis Index (WOMAC) scores similarly to naproxen. In paracetamol-treated patients, the mean changes from baseline to outcome in WOMAC pain, stiffness, and physical function scores were −21.6, −20.6, and −18.9 points, respectively [85]. In a Cochrane review of 15 RCTs involving 5986 participants, paracetamol was superior to placebo in five out of seven RCTs with a 5% relative improvement from baseline and a number-needed-to-treat of four to 16 to achieve a pain improvement; treatment with paracetamol did not modify WOMAC outcome [86].

The group of interest of the Italian Society of Anesthesia, Reanimation, and Intensive Care supported the use of paracetamol for chronic non-cancer pain [87]. In a meta-analysis of 15 RCTs, including 5133 patients of whom 3275 received an NSAID and 1858 paracetamol, paracetamol seemed slightly less effective than NSAIDs when larger numbers of patients with hip OA were included [88]. However, paracetamol was consistently associated with a better gastrointestinal safety profile than NSAIDs [88].

In his first meta-analysis of 10 RCTs, including 1712 young patients suffering from knee or knee and hip OA, Zhang et al. reported a significant pain reduction by paracetamol [89]. If initial treatment with paracetamol was favorable, paracetamol was recommended for continued long-term use for OA and lower back pain (LBP) [90,91]. For the management of knee OA, the European League against Rheumatism (EULAR) recommended paracetamol as the first and preferred long-term oral analgesic (Table 2) [92]. The European League of Associations of Rheumatology recommended paracetamol as the first-line treatment for mild-to-moderate OA pain because of its safety and effectiveness. NSAIDs should be considered in patients not responding to paracetamol and should be prescribed at the lowest effective dose and for the shortest duration [93]. The American College of Rheumatology for the treatment of hip and knee OA proposes a treatment with paracetamol before NSAIDs [94]. In its 2014 Osteoarthritis Care and Management Clinical Guideline, the National Institute for Health and Care Excellent (NICE) acknowledges a reduced effectiveness of paracetamol. NICE recommends that paracetamol be considered for pain relief and core treatments ahead of oral NSAIDs [95]. After having critically reviewed 1287 publications yielding 17 publications representing the best available evidence for analysis on the efficacy and safety of paracetamol for total joint arthroplasty, the American Association of Hip and Knee Surgeons, American Academy of Orthopedic Surgeons (AAOS), Hip Society, Knee Society, and American Society of Regional Anesthesia and Pain Managements supports with strong evidence the perioperative use of oral and intravenous paracetamol as a non-opioid adjunct for pain management of patients undergoing primary total joint arthroplasty both during hospitalization and following discharge [96].

In a subsequent analysis of 64 systematic reviews and 266 RCTs, Zhang et al. found that the analgesic effect size (i.e., mean effect of paracetamol group-mean effect of placebo group/standard deviation) of paracetamol was reduced not significantly from 0.21 to 0.14; however, the effect size was 0.1 and no longer significant when analyzing only high-quality-graded trials [97]. Therefore, in the recently updated of the Osteoarthritis Research Society International (OARSI) guidelines for non-surgical management of OA, paracetamol was conditionally not recommended; oral and transdermal opioids were strongly not recommended as well, and oral NSAIDs were not endorsed for cardiovascular or frail patients [98]. A recent Cochrane systematic review of 10 placebo-controlled RCTs involving 3541 patients stated that paracetamol alone provided only small clinical benefits at 3 weeks to 3 month follow-up and no increased risk of adverse events overall [99]. Nevertheless, the recent guidelines of the American College of Rheumatology/Arthritis conditionally recommend paracetamol for hand, hip, or knee OA at a maximum dose of 3 g/day, especially for patients with reduced therapeutic options because of contraindications to NSAIDs [100].

However, guideline reviews primarily focus on the analgesic efficacy in young patients and pay less attention to potential adverse events especially in the much less studied older patient population [101]. Nevertheless, when potential benefits and risks of harm are considered, paracetamol continues to be investigated and recommended as a first-line analgesic for older adults with mild-to-moderate pain [84,101]. In a systematic review of recommendations and guidelines for managing OA pain, the Chronic Osteoarthritis Management Initiative of the U.S. Bone and Joint Initiative, paracetamol was the first-line treatment across guidelines [90]. Although it reports findings questioning its efficacy, the NICE eventually recommended that paracetamol and/or topical NSAIDs to be kept in the NICE 2014 guidelines for OA and to be considered before oral NSAIDs, COX-2 inhibitors, or opioids [102]. The British Royal College of General Practitioners, the Primary Care Rheumatology Society, and the British Society for Rheumatology raised concerns about removing paracetamol from the recommended analgesics could result in unforeseen consequences from increased use of NSAIDs and opiates [102]. Makris et al. carried out a clinical review based on graded evidence from 92 studies on pharmacological and nonpharmacologic interventions, which were mostly primarily focused on older adults with OA. Given the scarcity of RCTs on older adults, the authors included reviews, guidelines, and consensus statements and concluded supporting a stepwise approach with paracetamol as first-line therapy for pain in the elderly [103]. The American Geriatric Society recommends extreme caution in the use of NSAIDs in the elderly because of frequent adverse events on the gastrointestinal, cardiovascular, and renal systems and the increased risk of hospitalization, renal toxicity, myocardial infarction, stroke, and death [104]; they recommend paracetamol as initial pharmacological therapy in the treatment of persistent pain, particularly MSP, because of its efficacy and safety profile (high quality of evidence; strong recommendation) [105]. They considered a liver failure as an absolute contraindication (high quality of evidence, strong recommendation) and hepatic insufficiency, chronic alcohol abuse, or dependence as relative contraindications (moderate quality of evidence, strong recommendation) [105]. The British Geriatric Society and British Pain Society indicate in the first UK guideline paracetamol as first-line treatment in older patients, particularly for MSP [106]. The American Geriatric Society recommends paracetamol as a first-line treatment for persistent pain with a 50–75% dose reduction in patients with liver failure [105]. Using a modified Delphi technique (two-round), a Brazilian expert panel suggested paracetamol as a possible alternative medication for pain in the elderly [107].

Interestingly, paracetamol is considered and commonly prescribed as first-line treatment for pain also in patients with dementia [108,109,110,111]. In observational studies, paracetamol improved pain and behavioral symptoms, enabling a reduction in psychotropic drugs in patients with dementia; in an RCT, paracetamol improved social engagement but did not affect behavioral symptoms [110]. These benefits were confirmed in an RCT on nursing home patients with mild-to-moderate dementia; using a stepwise approach, pain treatment with paracetamol improved daily living [111]. In a 6-month study by the Mount Sinai Visiting Doctors who provide home-based primary and palliative care to home-bound patients >80 years, the prescription for patients with moderate-to-severe pain of opiates and paracetamol increased from baseline to the end of the study from 48% to 57% and 52% to 91%, respectively [112].

Increasing or maintaining motor activity may be protective against OA progression and the excess mortality associated with OA [46]. Aging entails a progressive loss of muscle volume and function that impacts joint stability and motility and favors cartilage degeneration [113]. In this regard, paracetamol may be particularly useful. Carroll et al. have demonstrated that a 3-month resistance training and treatment with paracetamol 4 g/day versus placebo increased the cross-sectional area, deformation, and strain of the patellar tendon and decreased stiffness and modulus [114]. In a second RCT in older adults (mean ages 67 ± 2 and 64 ± 1 years), in comparison to the placebo, paracetamol 4 g/day or ibuprofen 1.2 g/day given during a 12-week knee extensor resistance training increases muscle and strength without altering muscle concentrations of muscle proteins, water, myosin heavy chain, and COX-1 and -2 [115].

Chronic low back pain (LBP) has a special place in pain-related disability; it is a complex syndrome resulting part from spinal OA and part from abnormalities of intervertebral discs and other structures [116,117]. Approximately 90% of all patients with LBP have non-specific LBP, which is a diagnosis based on the exclusion of specific causes including lumbar disc herniation, facet joint syndrome, lumbar spinal stenosis, spondylolisthesis, cancer, and infection. LBP is the first cause of disability worldwide [70,116,117,118,119]. Although paracetamol has long been recommended as the first-line treatment, recent systematic reviews found insufficient evidence to support its use in LBP [70,116,117,118,119]. Most recent guidelines advise against its use in chronic LBP (Table 2). Still, Morlion et al. recommend paracetamol as a first-line treatment for LBP in patients of advanced age or with gastrointestinal, cardiovascular, or renal comorbidities [117]. Koes et al. rightfully pointed out that analyzing the same evidence, paracetamol was recommended in three out of eight and was not recommended in four out of eight national guidelines for LBP [116]. Due to its safety, the Evidence-Based Medicine guidelines indicate paracetamol as a first-line analgesic for LBP [119].

The 2016 ASAS-EULAR guidelines recommend paracetamol for patients with axial spondylarthritis, an autoimmune disease of the spine, and who present failure, contraindications, or poor tolerance to first-line recommended treatments [120].

### 5.3. Cancer Pain

The World Health Organization developed a three-step analgesic ladder in 1986, specifically for cancer pain. Paracetamol and NSAIDs have been included in the first step of the ladder and the World Health Organization Model List of Essential Medicines [121]. Although paracetamol is rarely sufficient to control severe pain when administered in monotherapy, it is useful in the initial phase of treatment of cancer pain. According to Portenoy and Ahmed, paracetamol is “a reasonable alternative to a low dose of a strong opioid alone for opioid-naïve patients with moderate to severe cancer pain” [122]; these authors also suggest to use a fixed-dose of paracetamol in patients who began opioid therapy and are sensitive to their side effects. In the setting of multimodal analgesia, paracetamol has been administered with different opioids (i.e., codeine, hydromorphone, hydrocodone) and has consistently demonstrated an opioid-sparing effect on the postoperative pain [123]. Finally, in combination with weak or strong opioids, paracetamol provided clinically meaningful relief in patients with moderate or severe cancer pain [122,123]. Nevertheless, a 2017 Cochrane review of three RCTs comparing paracetamol alone with placebo or paracetamol plus an opioid with the same dose of opioid alone concluded that there was a high risk of bias and no high-quality evidence to support or refute the use of paracetamol in patients with cancer pain of any intensity [124]. Therefore, the European Society of Medical Oncology stated that there is no reason to recommend or refute paracetamol for mild-to-moderate cancer pain [125]. Conversely, a 2019 network meta-analysis of analgesic drugs for chronic cancer pain concluded that the top-ranking drug classes for global efficacy were non-opioid analgesics, including paracetamol, as well as NSAIDs and opioids [126]. It is considered the first-choice drug for mild-to-moderate pain in older patients with cancer naïve to opioids [127].

The American Society of Clinical Oncology (ASCO) indicates that paracetamol may be prescribed to relieve chronic pain and improve function in cancer survivors in whom there are no contraindications including dangerous drug–drug interaction (evidence quality: intermediate; strength of recommendation: moderate) [128]. The Spanish Oncological Society recommends paracetamol for pain management in cancer patients regardless of pain intensity and provided it is not contraindicated (level of evidence I, degree of recommendation A) [129]. The combination of paracetamol with strong opioids improves pain management and the sense of well-being [129].

## 6. Safety and Toxicity

### 6.1. Pharmacoepidemiology 

Oral formulations of paracetamol have been used for over 140 years with no clinically relevant adverse effects being usually apparent with recommended doses (i.e., up to 4 g per day) [130]. Almost all RCTs and meta-analyses reported numbers of adverse events from paracetamol that were inferior to those of NSAIDs and comparable to those of placebo. However, reviews on long-term observational data reported increased cardiovascular, gastrointestinal, and renal adverse events during therapy with paracetamol, especially in the high dose range; cases of acute liver failure have been reported after accidental and unintentional overdose of paracetamol [10,131,132,133,134,135,136,137,138,139,140,141,142,143,144]. Acute liver failure is infrequent with an approximate incidence for all causes of 1/million/year and is declining [134]. In recent Canadian reports, the numbers of paracetamol ingestions reporting to the Emergency Department declined from April 2011 to February 2019 [134]. Despite its rarity, acute liver failure generates interest and study by multiple disciplines because it affects all organ systems and requires substantial resource use [134]. Thusius et al. reported that the vast majority of patients survived and recovered without lasting medical sequelae. The liver transplant rate was 1.5% and the death rate <1%; the majority of both intentional and unintentional overdose patients underwent inpatient medical and psychiatric treatment [135]. Furthermore, drug-induced liver injury is a diagnosis of exclusion that has been ascribed almost only to paracetamol [143]. However, a recent review indicates that a drug-induced liver injury is idiosyncratic, unpredictable, and related to herbal and dietary supplements and to a number of many potentially toxic drugs to the liver [144].

In the patients enrolled in the US NIH registry of the Liver Failure Study Group between 2007 and 2013, acute liver toxicity from paracetamol overdosing was mostly related to psychiatric comorbidity including depression, frequent substance abuse, and increased impulsivity [132]. Psychiatric patients may require special monitoring of paracetamol consumption as they do with any medication. Patients who are fasting, consuming excessive alcohol drink, or suffering from liver diseases (i.e., non-alcoholic fatty liver diseases) may present a higher risk of toxic liver damage [132]. A cautious dose reduction to 2 g/24 h is advised for malnourished patients (weight < 50 kg) when paracetamol is used regularly in these patients. Not all initial safety warnings are being confirmed. For example, it has been suggested that paracetamol may alter the therapeutic response to conventional antidepressants [133]. A recent investigation in bipolar disorders indicated instead that paracetamol has no detrimental effect on affective disorder treatment; more specifically, it does not modify the lithium- or quetiapine-based bipolar disorder mood-stabilizing treatment outcomes [133].

### 6.2. Potential Mechanisms of Paracetamol Toxicity

Paracetamol overdose can cause liver damage and failure for which different mechanisms have been suggested [136,137,138]. At therapeutic doses, paracetamol is metabolized in the liver primarily by glucuronidation (50–60%) and sulfonation pathways (25–30%) and less by oxidation by cytochrome P450 2E1 (<10%). The latter produces only small amounts of the toxic metabolite NAPQI, which is detoxified by glutathione conjugation [137]. In paracetamol overdose or when sulfate availability and/or activity of sulfotransferases are low, NAPQI is formed in excess [135]. It binds to mitochondrial proteins, forming cytotoxic adducts, leading to mitochondrial dysfunction and severe hepatocellular necrosis [137]. Following the direct hepatotoxic effects of NAPQI, the inflammatory reaction and other critical events contribute to the evolution toward liver regeneration or liver irreversible damage. They include oxidative stress, reactive nitrogen formation, and JNK activation [138]. A key role is played by platelets that recruit macrophages and neutrophils; it has recently been shown that blocking the platelets C-type lectin-like receptor improves liver regeneration [136].

### 6.3. Treatments of Paracetamol Toxicity

N-acetyl cysteine has been widely used as a glutathione regenerator to treat paracetamol overdoses, and several other molecules are under investigation to treat paracetamol-induced liver damage [139,140,141].

Guidelines for paracetamol overdose include charcoal for patients that can be treated early after ingestion and intravenous N-acetyl cysteine for patients at risk of hepatotoxicity [141]. In a RCT on 24 healthy volunteers given paracetamol (1 g × 4 daily × 4 days) with N-acetyl cysteine or with placebo, N-acetyl cysteine prevented glutathione depletion without interfering with paracetamol analgesia [141].

## 7. Recent Findings and Future Directions

The numbers of studies, systemic reviews, and meta-analyses on paracetamol is high and growing. However, large, authoritative clinical trials are costly, and it is unclear whether they will be done [145,146].

Recently, Abdel Shaheed analyzed data from 36 publications and reported that paracetamol produced modest pain relief in hip and knee osteoarthritis, tension headache, and post-craniotomy pain and that it was ineffective in other conditions [145]; the authors concluded with the need for large, high-quality trials to reduce uncertainty about the efficacy of paracetamol in common pain conditions. In the last few months, new published studies report the safe use of paracetamol in patients with chronic kidney disease [147], the significant efficacy of paracetamol for post-arthroplasty pain [148], the lesser efficacy of paracetamol than ibuprofen for post-laminectomy pain [149], the opioid-sparing effects of paracetamol in the Emergency Department [150], the recommendations against paracetamol in lumbar spinal stenosis with neurogenic claudication [151], and the recommendations in favor of paracetamol for postcesarean pain [152] and for migraine [153,154].

RCTs and other evidence should be analyzed critically by experts before adopting (or refusing) the results to daily clinical practice [155]. Clinical guidelines fill the gap between scientific evidence and its application in clinical practice. They have the advantage of being more comprehensive because they are generated from the consolidated medical and because they consider a number several economical, epidemiological, research, and therapeutic factors including, for example, the availability of alternative treatments. In our opinion, practice guidelines are essential tools in managing the complexity of decision-making processes.

## 8. Conclusions

Paracetamol has been one of the most recognizable drugs, both on- and off-prescription, and it is likely to remain so in the future. As a result of the global aging, painful and disabling conditions are increasing. Paracetamol has a favorable safety profile that will be of utmost importance across all ages and especially in the elderly. Liver toxicity is a concern, but it is questionable at doses up to 4 g/day. Although it underwent intense scrutiny of its efficacy and safety, paracetamol is recommended by guidelines for the treatment in diverse acute and chronic pain.

Finally, the safety and tolerability record and the safety advantages of paracetamol over other classes of NSAIDs and opioid analgesics have been among the reasons for its maintenance or inclusion in pain treatment guidelines by expert panels.

Notably, several reports support the equivalent safety and efficacy profile of both oral and intravenous administrations. This conclusion is particularly relevant, suggesting that paracetamol may be used equally through both administration routes in several clinical cases in the emergency setting in chronic pain conditions.

## Figures and Tables

**Figure 1 jcm-10-03420-f001:**
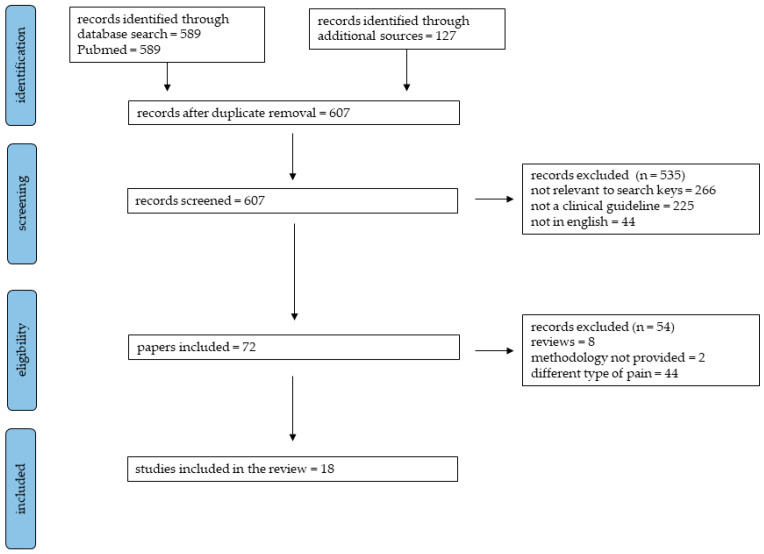
Flowchart of study screening, eligibility and inclusion.

**Figure 2 jcm-10-03420-f002:**
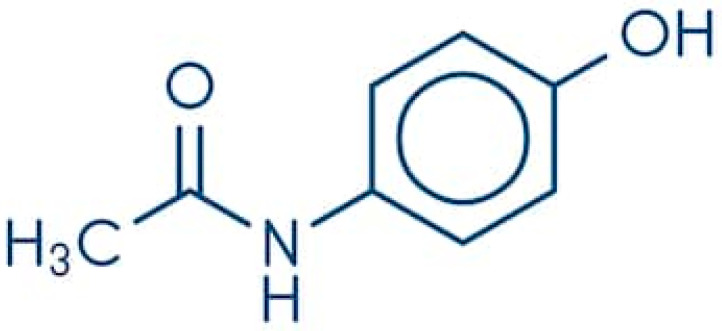
Molecular structure of paracetamol (acetaminophen).

**Table 1 jcm-10-03420-t001:** Clinical pharmacological activities of paracetamol.

analgesic	high activity
antipyretic	high activity
anti-inflammatory	low activity
antiplatelet	low activity
antidepressant	anecdotal
cognitive-enhancer	anecdotal

**Table 2 jcm-10-03420-t002:** Guideline assessment according to the AGREE II instrument.

Organization/Society	Condition	AGREE II Domain Score (%)						
		Scope and Purpose	Stakeholder Involvement	Rigor of Development	Clarity of Presentation	Applicability	Editorial Independence	Overall
ASAS-EULAR	axSpA	97	65	67	91	30	58	68
OPTIMa	LBP	100	77	73	86	14	67	70
NICE	LBP	100	92	89	68	42	90	80
EBM	LBP	97	49	46	94	35	21	57
ACR	OA	97	89	64	89	27	62	61
AAOS/ASRA	OA	89	42	34	69	0	52	51
ESCEO	OA	100	66	64	100	24	47	67
NICE	OA	100	99	83	100	86	87	93
OARSI	OA	100	77	54	93	40	92	76
ACP/ASIM	MSP	100	86	78	97	12	100	79
AGS	MSP	87	57	34	94	1	48	56
BGS, BPS	MSP	100	71	60	94	16	33	63
AHS	HA	94	71	69	94	12	92	72
EFNS	HA	89	69	39	100	5	50	59
NICE	HA	89	89	73	86	42	90	78
ASCO	CP	100	80	86	100	51	75	82
ESMO	CP	94	71	83	86	6	54	59

Abbreviations: ACP-ASIM: American College of Physicians–American Society of Internal Medicine; ACR: American College of Rheumathology; AAOS: American Association of Orthopedic Surgeons; AGS: American Geriatric Society AHS: American Headache Society; ASAS-EULAR: Assessment of SpondyloArthritis international Society; ASCO: American Society of Clinical Oncology; ASRA: American Society of Regional Anesthesia and Pain Medicine; axSpA: axial spondyloarthritis; BGS: British Geriatric Society; BPS: British Pain Society; CP: cancer pain; EBM: Evidence-Based Medicine; EFNS: European Federation of Neurological Societies; ESCEO: European Society for Clinical and Economic Aspects of Osteoporosis, Osteoarthritis and Musculoskeletal Diseases European Society for Clinical and Economic Aspects of Osteoporosis, Osteoarthritis and Musculoskeletal Diseases; ESMO: European Society of Medical Oncology; EULAR: European League Against Rheumatism; HA, headache; LBP: low back pain; MSP: musculoskeletal pain; OA: osteoarthritis; NICE: National Institute for Health and Care Excellence; OARSI: Osteoarthritis Research Society International; OPTIMa: Ontario Protocol for Traffic Injury Management.

**Table 3 jcm-10-03420-t003:** Guideline recommendations for pain treatment with paracetamol.

Organization/Society	First Author, Year	Condition	Recommendation	Comments
**ASAS-EULAR**	van der Heijde, 2016	axSpA	R	to be considered after NSAIDs failed
**OPTIMa**	Wong, 2016	LBP	R	recommended in acute LBP
**NICE**	NICE 2020	LBP	CR	not recommended alone, recommended in association with opioids
**EBM**	EBM 2019	LBP	R	recommended for acute and chronic LBP
**ACR**	Kolasinski, 2020	OA	CR	recommended for patients intolerant to NSAIDs, monitor liver function
**AAOS/ASRA**	Fillingham, 2020	OA	R	
**ESCEO**	Bruyere, 2014	OA	R	first line for short-term treatment (<3 g/day); not for long-term treatment
**NICE**	NICE 2020	OA	R	to be considered ahead of NSAIDs
**OARSI**	Bannuru, 2019	OA	CNR	
**ACP/AAFP**	Qasem, 2020	MSP	CR	
**AGS**	AGS Panel, 2009	MSP	R	contraindicated in liver failure; not exceed max 4 g/day dose
**BGS, BPS**	Abdulla, 2013	MSP	R	elderly population; not to exceed max 4 g/day dose

**AHS**	Marmura, 2018	HA	R	
**EFNS**	Bendtsen, 2010	HA	R	1 g for acute therapy
**NICE**	NICE, 2021	HA	R	indicated for migraine and tension headache
**ASCO**	Paice, 2016	CP	R	avoid drug interaction
**ESMO**	Fallon, 2018	CP	NR	
**WHO**	WHO, 2019	pain	R	

Recommendations by medical societies for pain management with paracetamol in different clinical conditions. Each society followed specific criteria to grade quality evidence and recommendations that should be consulted. Abbreviations: same as in Table 1. Recommendations: light red-CNR: conditionally not recommended (probably ineffective); light green-CR: conditionally recommended (effective in some patients); yellow-NR: no recommendation (not enough reason to support or refute its use); green-R: strongly recommended or recommended (effective or probably effective).

## Data Availability

Not applicable.

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
