# Peer review of "Paracetamol: A Review of Guideline Recommendations"

_jcm, 2021, doi:10.3390/jcm10153420_

Round 1

Reviewer 1 Report

The article is a narrative review of some of the evidence used as the basis of various guidelines for conditions that are both chronic, non-malignant, and malignancy.  There are several problems with the author’s approach.

  1. There is not a thorough review of each guideline; the evidence for the recommendations; and the strength of these recommendations. For example, in Table 2, the ASCO guidelines for the use of acetaminophen are labeled Expert Opinion as the basis for the recommendations, but in the text the recommendations from the guidelines are scored as moderate with intermediate evidence.  Guidelines for various conditions that are done by expert opinion are lumped with those that used a defined and more rigorous methodology and seemingly given equal weight in the conclusions.
  2. Methodology exists in the literature to rigorously carry out a review of reviews or meta-analyses.1, 2 For example, Nguyen articulated one set of methods:  (1) formulating a focused and clinically-relevant question; (2) designing a detailed review protocol with explicit inclusion and exclusion criteria; (3) performing a systematic literature search of multiple databases and unpublished data, in consultation with a medical librarian, to identify relevant studies; (4) meticulous data abstraction by at least two sets of investigators independently; (5) assessing risk of bias in individual studies; (6) quantitative synthesis with meta-analysis; and (7) critically and transparently ascertaining quality of evidence.2
  3. The authors have not incorporated these methods into this review, seeking only to carry out a less rigorous narrative review of some, but not all, the literature as the basis for each guideline. One of the references which discusses the use of acetaminophen for OA has as its first author an employee of the pharmaceutical firm that manufactures Tylenol ® in the United States. 3
  4. The authors conclude that deaths due to acetaminophen’s hepatic toxicity is diminishing based on a single report from Canada. Numerous other contemporary articles describe this problem as an ongoing issue—500 deaths yearly in the United States; continued deaths due to liver toxicity from this drug in Europe.4  Their final conclusion that this is a drug with an excellent safety profile is misleading and un-referenced.  There is a narrow window of 1-3 or 4 grams daily before liver toxicity can be seen (biochemical depletion of protective enzymes begins at 2 grams).  The use of the drug in the elderly lacks a large body of evidence.  The ongoing opioid epidemic in the United States (referred to in one place in the article as a “so-called epidemic”) increases the likelihood of this drug being used but the safety and conflicting evidence of efficacy do not render the drug an acceptable alternative to opioids simply because there are no other classes of analgesia available, which seems to be an underlying theme in the article.
  5. The authors discuss use of the “fifth vital sign” in the United States as one of the causes of overuse of opioids but do not address the overuse of acetaminophen that has occurred since the 1980’s in the United States and now in Europe.
  6. There are numerous grammatical and spelling errors in the English used, especially in the first part of the article. For example, Cochrane reviews are referred to variably as Cochrane or Cochrane’s.  On line 662, regeneration is misspelled.  The latter part of the article appears to be better in terms of spelling and grammar.  If this article were to be resubmitted, either a translation service needs to be used, or the author of the latter half of the article needs to be asked to proof the article carefully. 
  7. Table 2 has some, but not all, of the abbreviations used in the footnotes. The recommendations for use include “CR” which is not explained anywhere in the footnote or text.  There is a symbol used prior to the numbers (see line 238) that I am not familiar with and am unable to locate and needs clarification.  The description of types of nerve fibers appears to describe the Aδ fiber but the symbol is not clear.
  8. Most, but not all the references (e.g., Reference 61) lack a “doi” identifier. The references need to be carefully reviewed for omissions.   

Bibliography

  1. Aromataris E, Fernandez R, Godfrey CM, Holly C, Khalil H, Tungpunkom P. Summarizing systematic reviews: methodological development, conduct and reporting of an umbrella review approach. Int J Evid Based Healthc. Sep 2015;13(3):132-40. doi:10.1097/xeb.0000000000000055
  2. Nguyen NH, Singh S. A Primer on Systematic Reviews and Meta-Analyses. Semin Liver Dis. May 2018;38(2):103-111. doi:10.1055/s-0038-1655776
  3. Temple AR, Benson GD, Zinsenheim JR, Schweinle JE. Multicenter, randomized, double-blind, active-controlled, parallel-group trial of the long-term (6-12 months) safety of acetaminophen in adult patients with osteoarthritis. Clin Ther. Feb 2006;28(2):222-35. doi:10.1016/j.clinthera.2006.02.004
  4. Lee WM. Acetaminophen (APAP) hepatotoxicity-Isn't it time for APAP to go away? J Hepatol. Dec 2017;67(6):1324-1331. doi:10.1016/j.jhep.2017.07.005

Author Response

Dear Reviewer,

Following are our responses to specific points raised in the Review.

1. This narrative review is intended to provide health professionals and patients with authoritative guidelines for the treatment with paracetamol of most common pain conditions (ie, musculoskeletal pain, migraine, cancer pain) within the context of the current knowledge of pain and of the alternative treatments. The numbers of studies and reviews on paracetamol/acetaminophen is high and growing. Just in the few weeks since the original submission of this paper, one can find quite a few new studies reporting, for example, the safe use of paracetamol in patients with chronic kidney disease (Dolati et al., 2020), the lack of advantage of antiepileptics over paracetamol or NSAIDs for febrile seizures (Offringa et al., 2021), the significant efficacy of paracetamol for sore throat (Cimen et al., 2021) and for post-arthroplasty pain (Anger et al., 2021), a lesser efficacy of paracetamol than ibuprofen for post-laminectomy pain (Akbas et al., 2021), the opioid sparing effects of paracetamol in Emergency Department (Ramdin et al., 2021), recommendations against paracetamol in lumbar spinal stenosis with neurogenic claudication (Boussieres et al., 2021), recommendations in favor of paracetamol for postcesarean pain (Roofthooft et al., 2021) and for migraine (VanderPluym et al., 2021, JAMA; Robbins, 2021, JAMA), and the questionnable benefits of NSAIDs on postoperative pain and opioid dose reduction (VanderPluym et al., 2021).

However, we included guidelines and recommendations from highly cited, authoritative medical societies and agencies (AHS, BPS, NICE, WHO, CDC, EULAR, BGS, BPS, ASCO etc) and few, less authoritative, guidelines from national societies including from our own country; if the Reviewer feels it necessary, the latter can be removed.

ASCO panelists used specific methods and criteria to develop guidelines which include A Measurement Tool to Assess Systematic Reviews (AMSTAR), GuideLines Into Decision Support methodology (GLIDES), a 3-degree formal and informal consensus process (ie, strong-moderate- weak), and multistep revision processes (https://ascopubs.org/doi/suppl/10.1200/jco.2016.68.5206/ suppl_file/MS_2016.685206.pdf). Each ASCO recommendation has an accompanying statement including the one we reported in the text.

2. Quantitative reviews and reviews of reviews exist. For example, very recently, Shaheed analyzed data from 36 publications and reported that paracetamol produced modest pain relief in hip and knee osteoarthritis, tension headache and after craniotomy and was ineffective in other conditions (Shaheed et al., 2021); authors highlights the need for large, high quality trials to reduce uncertainty about the efficacy of paracetamol in common pain conditions. More quantitative meta-analyses are being planned (Stormholt et al., 2021). Our Institute routinely carries out meta-analyses (Vaschetto et al., 2021; Carron et al., 2020; Carron et al., 2021; De Cassai et al., 2021).

Although meta-analyses are powerful and informative, good clinical care requires also knowledge of the best available evidence whether it comes or not from them; further, in view of their proliferation, their validity has been questioned (Packer, 2017; Packer et al., 2019; Thornly et al., 2018). Clinical guidelines fill the gap between scientific evidence and its application in clinical practice. They have the advantage of being more comprehensive because they are generated from the consolidated medical experience of medical institutions or societies. Recommendations from guidelines take into account different economical, epidemiological, research and therapeutic factors including, for example, whether or not alternative treatments are available. While methods for grading the evidence and the strength of recommendations are pretty consistent, medical societies use more variable and complex methods to reach the consensus by panelists. For example, using a similar GRADE approach, voting procedure and consensus threshold, the European Society for Clinical and Economic Aspects of Osteoporosis, Osteoarthritis and Musculoskeletal Diseases (ESCEO) recommends low- dose, short- term acetaminophen, pharmaceutical grade glucosamine and chondroitin sulfate while the Osteoarthritis Research Society International (OARSI) strongly recommends against their use (Arden et al., 2021). Starting from the same evidence. Other societies use even more complex methods. For example, guidelines for postoperative pain of the American Society of Anesthesiologists (ASA) are developed with a seven-step procedure; step 1 is about consensus on evidence and step 2 literature selection; other are about survey and review of comments (ASA, 2012). Similarly, the Centers for Disease Control and Prevention, National Comprehensive Cancer Network, and ASCO publish clinical practice guidelines for the management of chronic pain that differ at least in the perception of the readers (Schatz et al., 2021).

An overview of which specific guideline does or does not recommend or recommends against paracetamol for a specific condition adds infos to physicians and patients.

3. Clinical trials are extremely expensive; for this reason, they are conducted in high-income countries and are linked, directly or indirectly, to the pharmaceutical industry (Baron et al., 2015; Egilman et al., 2019; Kellstein and Rina Leyva, 2020). This is an unfortunate risk of bias but is almost impossible to avoid especially when trials are carried out to seek approval of regulatory agencies such as FDA or EMEA. RCTs should be analyzed critically by experts before adopting (or refusing) the results to daily clinical practice (Mielke and Rohde, 2020). On the other hand, the consensus process of guidelines development may simply reinforce the biases of the assembled experts. For these reasons, several societies include methodologists, clinicians, and public representatives.

4. The incidence of severe toxicity (ie, hospitalization or death) of paracetamol varies across countries or regions. Lee et al. report impressive numbers of paracetamol toxic effects (ie, 100.000 calls, to US poison centers, 50.000 ED visits and 10.000 hospitalization). However, in multi-etnic Asia country, toxicity of paracetamol was low; mortality and morbidity approach 0 rate despite high doses of paracetamol ingestion and delayed presentations to hospital (Marzilawati et al., 2012). In our own experience (1200 bed university hospital), in 2019 we reported 32 cases of acute renal failure from chronic NSAIDs (7 with a permanent renal damage), 2 cases of gastric perforation from NSAIDs (1 fatal), 1 case acute liver failure from NSAIDs (nimesulide), 2 cases of opioid overdose and 0 cases of toxicity from paracetamol (Freo, XIV Course on Neuromodulation). Also, year deaths related to paracetamol seems to decline in England and Wales (https://www.statista.com/statistics/471227/death-by-paracetamol-drug-poisoning-in-england-and-wales/) and to increase in Australia (Cairns et al., 2019). These regional differences cannot be easily explained. They may result from undetermined genetic factors (Caparrotta et al., 2018) or from the way acetaminophen is available (ie, marketed) and prescribed. The picture is further complicated by the fact that drug-induced acute liver injury (DILI) remains a diagnosis of exclusion. Recently DILI has been linked not only to acetaminophen but, much more often, to several other liver toxic drugs, to dangerous drug-drug interaction and to herbal and dietary supplements (Lee BT et al, Current Gastroenterological Report, 2021), which probably have been overlooked in the past. Also, in a just published editorial, Professor Schatman points out that there is “a war” and “hyperbolically stating that any utilization of acetaminophen, even in a single instance, will result in grave hepatic injury” is “a disingenuous point” from physicians and researchers “suggesting that opioids are the only effective and viable treatment for chronic pain” (Schatman et al., 2021).

Further, the issue about safety of paracetamol should be put in the context of current pain treatments. Pain is the leading cause of disability involving billions of people worldwide (Cieza et al., 2020). Dr Lee reports 500 death/year in the US and that “APAP toxicity dwarfs all other prescription drugs”. This may be true for DILI; it is definitely not true for pain drugs as well as for other classes of medications. Just in 2019 in USA there were 70,630 deaths from opioid overdose (approximately 14,139 on prescription) with a nearly 5% increase from 2018 (https://www.drugabuse.gov/drug-topics/  trends-statistics/overdose-death-rates. NSAIDs cause 16,000 deaths per year and send 100,000 people to the emergency room in the U.S. (Krueger 2013https://bangordailynews.com/2015/07/27/health/why-the-fda-is-warning-americans-about-ibuprofen/).

Finally, rates of treatment interruptions from drug adverse effects are high for opioids (ie, 40-60%, Freo et al., 2021), substantial for NSAIDs (ie, 10-30%, Lisse et al., 2003; Day et al., 2000) and very low for acetaminophen. Although not a topic of this paper, in contrast to other pain medications paracetamol is commonly used in children and pregnant women. Therefore, taking all data into account, the safety profile of acetaminophen is, if not excellent, at least good or more favourable than those of other analgesics. In any event, we replaced excellent throughout the paper.

We do not think at all that paracetamol is an alternative to opioids (Freo et al., 2018, Freo et al., 2021); however, paracetamol may help to reduce the doses of opioids (Bettiol et al., 2021; Daniels et al., 2019; Gupta et al., 2016; CDC 2016) and can be of interest to all those patients in whom NSAIDs and opiates are contraindicated. We agree that paracetamol lacks a large body of evidence in the elderly; however, this is true for any class of analgesics (Freo et al., 2021). Sadly, although the very idea of aging has been revised (ie, youngest-, middle- and oldest-old) and the elderly are the fastest growing portion of population suffering from pain, clinical pharmacological research on pain has focused on relatively younger patients. This is mentioned in the paper and is probably due to the risk (fear) of  failure of (very expensive) trials. Finally, although several new potential targets for pain treatment have been discovered (Ciotu and Fischer, 2020), advanced human trials have been fairly disappointing.

The American Geriatric Society and British Geriatric Society support the use of paracetamol as first line for pain control in the elderly. NSAIDs and opioids are definitely more dangerous in aged than in young people and increase gastroenterological, renal and vascular morbidity and mortality. Some 20 years ago, the COX inhibitor rofecoxib was withdrawn for similar reasons. In contrast to dr Lee, we think that none of pain drugs (ie, paracetamol, NSAIDs, opioids, SNRI and GABA drugs) “should go away”. It should be recognized, however, that available pain drugs are not highly effective and only a portion of patients achieve a significant relief. Pain remains an unmet need, worldwide. In our opinion, until new agents will be approved, the available analgesics should be used at their best under the guidance of experts’ guidelines.

5. Please see also above. Rates acute liver failure from acetaminophen varies with countries (Gulmez et al., 2015); specifically, it is not common in Germany (Hadem et al., 2012), France (Girard et al., 2019), Spain (Escorsell et al., 2007), Greece, Italy and Portugal (Gulmez et al., 2015); in Switzerland Martinez-De la Torre reported an increase in calls from acetaminophen poisoning after 1000 mg tablets became available; interestingly there was no increase in death rate (Martinez-De la Torre e 2020).

6. We checked spelling and grammar throughout. We changed Cochrane’s to Cochrane throughout and corrected regeneration.

7. CR stands for conditionally recommended and was already in the footnote.

8. We added the DOI identifiers to the references.

The reference list has been largely altered. As changes make the list poorly readable, we decide to do NOT indicate them in the revised version. However, the file with changes on the original reference list is available upon request of the Editors or Reviewers

Reviewer 2 Report

Overall very good and comprehensive work. Line 268: Please add 24 hours. Line 372 and 378: Please add the references`numbers. Line 402: Please delete "intravenous". Line 431: Please add the number of the reference. Moore RA et al. Single dose oral analgesics for acute postoperative pain in adults - an overview of Cochrane reviews. Cochrane Database of Systematic Reviews 2015, Issue 9. Art. No.: CD008659. DOI: 10.1002/14651858.CD008659.pub3, and Moore RA et al. Non-prescription (OTC) oral analgesics for acute pain - an overview of Cochrane reviews. Cochrane Database of Systematic Reviews 2015, Issue 11. Art. No.: CD010794. DOI: 10.1002/14651858.CD010794.pub2, both, should be mentioned. These reviews showed superiority of the combination of paracetamol and ibuprofen compared to other compounds, and furthermore, an excellent safety profile. No increase in cardiovascular risk is seen with ibuprofen at doses up to 1,200 mg per day, which is the highest dose generally used for over-the-counter (OTC) preparations taken by mouth in the European Union (EU). (https://www.ema.europa.eu/en/news/prac-recommends-updating-advice-use-high-dose-ibuprofen). In addition, at least one article showed more anti-inflammatory activity by using the combination of paracetamol and EPA/DHA (Caughey GE et al. Complement Ther Med 2010;18:171-4).

Author Response

Thanks for the indications.

The two Cochrane reviews have now been included as well as the paper on the synergistic effect of EPA/DHA on anti-inflammatory activities of paracetamol.

Round 2

Reviewer 1 Report

  1. This paper is a lengthy narrative of the guidelines in the literature without any critical look at the quality of each guideline. The authors justify this approach by indicating that, in essence, clinical judgement is as important as the evidence.   Multiple conflicting reports are cited without any effort to critique the quality of the report.  For example, on line 380, the authors assert that the drug has shown superiority without any references and then assert on line 381 that it failed to show superiority, again without any references.

In their response, they provide some justification for this approach and this discussion in item 2 should be incorporated into the discussion of how they approached the validation of each guideline. 

References would also be helpful for lines 380 and 381, respectively. 

  1. The authors minimize the 500 deaths yearly in the United States from paracetamol induced liver injury by declaring that pain is a global problem and 17000 deaths from opioids (from any etiology) yearly in the United States are of more concern. The 500 deaths may in part be prevented by attention to the published guidelines on the safe use of this drug; in contrast to a significant number of the deaths due to opioids related to overdose, use of illicit opioids, etc. 
  2. There are continued grammatical errors, e.g., on line 95—"…without several (needs preposition--of) the side effects…”; and line 122—"At difference from with… “ This clause might be better expressed as (in contrast to other).

Author Response

Dear Reviewer,

we altered the paper according to indications.

References have been included at lines 380-381 (currently lines 339-340). Previously, references were present in the list and were mentioned but not discussed here.

We do not intend to dismiss the high numbers of deaths from paracetamol in the US. For this reason we included dr WM Lee paper on paracetamol with his provocative title of dropping paracetamol from the therapeutic armamentarium. However, we are not sure how poisoning data from US and Australia can ben transferred to other coutries where the rates of poisoning are low or almost 0.  Drug-induced liver failure is a diagnosis of exclusion and its precision may be low; in addition, other drugs and nutrients can be, either alone or in combination, toxic to the liver (the paper by BT Lee et al has been included) and the way they are available, prescribed or used in different fashions across countries. The same may hold true for protective factors.